# Low-Dose Aspirin after ASPRE—More Questions Than Answers? Current International Approach after PE Screening in the First Trimester

**DOI:** 10.3390/biomedicines11061495

**Published:** 2023-05-23

**Authors:** Piotr Tousty, Magda Fraszczyk-Tousty, Sylwia Dzidek, Hanna Jasiak-Jóźwik, Kaja Michalczyk, Ewa Kwiatkowska, Aneta Cymbaluk-Płoska, Andrzej Torbé, Sebastian Kwiatkowski

**Affiliations:** 1Department of Gynecology and Obstetrics, Pomeranian Medical University, 70-111 Szczecin, Poland; 2Department of Neonatology and Neonatal Intensive Care, Pomeranian Medical University, 70-111 Szczecin, Poland; 3Department of Gynecological Surgery and Gynecological Oncology of Adults and Adolescents, Pomeranian Medical University, 70-111 Szczecin, Poland; 4Department of Nephrology, Transplantology and Internal Medicine, Pomeranian Medical University, 70-111 Szczecin, Poland; 5Department of Reconstructive Surgery and Gynecological Oncology, Pomeranian Medical University, 70-111 Szczecin, Poland

**Keywords:** preeclampsia, prenatal screening, first trimester, aspirin, ASPRE

## Abstract

Preeclampsia (PE) is a multi-factorial disorder of pregnancy, and it continues to be one of the leading causes of fetal and maternal morbidity and mortality worldwide. Aspirin is universally recommended for high-risk women to reduce preeclampsia risk. The purpose of this review is to summarize the recommendations of various scientific societies on predicting preeclampsia and their indications for the inclusion of acetylsalicylic acid (ASA) prophylaxis. Fourteen guidelines were compared. The recommended dose, screening method, and gestational age at the start of the test vary depending on the recommendation. The societies are inclined to recommend using increasingly higher doses (>75 mg) of ASA, with many encouraging doses from 100 mg upward. Most societies indicate that the optimal time for implementing aspirin is prior to 16 weeks’ gestation. Following the publication of the Aspirin for Evidence-Based Preeclampsia Prevention (ASPRE) trial results and other papers evaluating the Fetal Medicine Foundation (FMF) screening model, a large number of societies have changed their recommendations from those based on risk factors alone to the ones based on the risk assessment proposed by the FMF. This allows for the detection of a high-risk pregnancy population in whom aspirin will be remarkably effective in preventing preterm PE, thereby decreasing maternal and fetal morbidity.

## 1. Introduction

Preeclampsia (PE) is a multi-factorial disorder found in 2–8% of pregnancies. It, unfortunately, continues to be one of the leading causes of fetal and maternal morbidity and mortality worldwide, accounting for more than 70,000 maternal deaths every year [1,2]. As defined by the International Society for the Study of Hypertension in Pregnancy (ISSHP) in 2021, gestational hypertension is accompanied by one or more of the following new-onset conditions at ≥20 weeks’ gestation:

1.Proteinuria.2.Other maternal end-organ dysfunction, including:
-neurological complications (blindness, stroke, paresis, severe headaches, persistent visual scotomata);-pulmonary edema;-hematological complications (e.g., platelet count < 150,000/μL, DIC, hemolysis);-acute kidney injury (AKI) (such as creatinine ≥ 90 μmol/L or 1 mg/dL);-elevated transaminases, such as alanine transaminase (ALT) or aspartate transaminase (AST) > 40 IU/L), with or without right upper quadrant or epigastric abdominal pain.3.Uteroplacental dysfunction: placental abruption, angiogenic imbalance, fetal growth restriction (FGR), abnormal umbilical artery Doppler waveform analysis, or intrauterine fetal death [3].

Currently, the only method to treat a patient diagnosed with PE is to terminate the pregnancy. In PE cases diagnosed > 37 weeks’ gestation (term PE), such management does not pose a large challenge. When it comes to preterm PE or early-onset PE (<34 weeks’ gestation), however, we judge between the induced complications of preterm birth and the increased risk of maternal or fetal morbidity and mortality (resulting from, for example, placental abruption) [4]. In addition to the obvious early complications, preeclampsia carries the risk of long-term complications for both mothers and fetuses. Mothers are far more exposed to cardiovascular diseases, obesity, and kidney diseases. Children born to mothers with PE have increased risk of abnormal psychomotor development, insulin resistance, diabetes mellitus, and cardiovascular diseases [5,6,7,8,9]. The first paper showing the effects of acetylsalicylic acid (ASA) in preventing PE was published more than 30 years ago [10]. Hence the interest of numerous authors in the early detection of women at increased risk of developing PE, who may be the group with the greatest effectiveness of implemented ASA [11,12,13,14,15]. There have been many studies showing the effects of aspirin use, timing of treatment inclusion, and the selected population on the incidence of PE [16,17,18]. Although the effect of ASA on the prevention of PE occurrence seems already proven, there is still no international consensus on several controversial issues:Choice of the optimal screening method in the first trimester.Selection of an appropriate cut-off point for selected populations at high risk of developing PE.Selection of the appropriate dose of ASA.The timing of implementation and end of treatment.

The aim of this study was to look at the state of current knowledge on the prediction and prevention of PE with ASA. This article summarizes the recommendations of various scientific societies for predicting PE and their indications for the implementation of ASA prophylaxis and looks at the changes following the publication of the Aspirin for Evidence-Based Preeclampsia Prevention (ASPRE) study in 2017 [19].

## 2. Search Strategy and Article Selection Criteria

PubMed, Web of Science, and Google Scholar were searched through to 31 March 2023, with the search terms “hypertensive disorders in pregnancy”, “preeclampsia”, and “hypertension in pregnancy”. We cross-listed these terms with the following: “aspirin”, “screening”, “prediction”, “prevention”, “management”, “guidelines”, and “society”. We focused on publications written after 2017.

## 3. An outline of the Pathophysiology of PE and the Mechanism of Action of ASA

PE is considered by researchers to be caused by an abnormal process of the so-called placentation. In a normal, physiological pregnancy, the invasion of trophoblast cells and the angiogenic (including vascular endothelial growth factor (VEGF) and placental growth factor (PlGF)) and immune factors secreted by them induce spiral artery remodeling. Such remodeling enables adequate perfusion of the arteries and, thus, normal placentation and development of pregnancy [20,21]. In the case of PE, this process is disturbed, and the first stage of the first trimester does not see normal trophoblast invasion. Several elements are involved, such as the aforementioned immune and angiogenic factors, genetic factors, and maternal diseases (e.g., pre-gestational diabetes mellitus, chronic hypertension) [21,22,23,24]. In the second stage, it leads to abnormal maternal–fetal perfusion. To date, abnormal maternal–fetal perfusion has been equated with subsequent placental hypoxia, although there are reports claiming that oxygenation in FGR and PE can be even higher than it is in normal pregnancies, whereas it is flow rates in these diseases that may be more relevant. Due to abnormal perfusion, the placenta generates oxidative stress leading to the release of inflammatory mediators and antiangiogenic factors into the maternal circulation. The latter two cause vascular endothelial damage, excessive platelet aggregation, and vasoconstriction through a decrease in nitric oxide (NO) synthesis, leading to the clinical manifestation of preeclampsia in the form of any of the disorders listed in the ISSHP definition [3,22,25,26,27,28,29,30,31]. What does the pathophysiology of preeclampsia have in common with the mechanism of action of aspirin?

Aspirin inhibits two cyclooxygenase isozymes (COX-1 and COX-2). Cyclooxygenases mediate in the production of prostanoids that include prostaglandins, prostacyclins, and thromboxanes [32,33]. Under normal conditions, COX-1 regulates prostacyclins and thromboxane in the vascular endothelium and platelets, where the former promote vasodilation and inhibit platelet aggregation, while thromboxane has the opposite effect [32,33,34]. COX-2, on the other hand, is mainly involved in regulating the inflammatory response by releasing prostaglandins—this cyclooxygenase isoform is inhibited by high-dose aspirin. Low-dose aspirin (LDA) mainly has an affinity for COX-1, causing an increase in the ratio of prostacyclins to thromboxane (in PE, the observations point to an exactly opposite situation). It also shows a slight anti-inflammatory effect [34,35,36,37]. Additional effects of LDA include immunomodulation, endothelial stabilization, influence on cytokine production, and inhibition of the production of anti-angiogenic factors such as soluble Fms-like tyrosine kinase-1 (sFlt-1) [38,39,40,41]. The latter element participates in inactivating VEGF and PlGF, which are responsible for supporting normal placentation. Given the multitude of mechanisms of action of LDA, its role in decreasing the incidence of PE is not fully known. On the other hand, proper spiral artery remodeling is known to be necessary to prevent PE. Therefore, aspirin prophylaxis should be implemented even before this process begins [39,42].

## 4. What Are the Benefits of ASA Prophylaxis?

Although the first reports on aspirin’s role in preventing preeclampsia were published as early as in the 1980s, this effect was only confirmed much later. In 2007, a meta-analysis evaluating 32,217 women with risk factors for preeclampsia showed a slight decrease in its incidence among those patients who received ASA (Relative risk (RR) 0.90, 95% Confidence interval (CI) 0.84–0.97). The meta-analysis consisted of various tests of aspirin dosage (50–150 mg) and the time of its inclusion (in many cases, ASA was included after 20 weeks’ gestation) [18]. In subsequent years, further meta-analyses showed that both the dose and the time of inclusion mattered [43,44,45]. One of such studies enrolled 11,348 women with risk factors for preeclampsia and showed that the incidence of PE (RR 0.47, 95% CI 0.34–0.65) and intrauterine growth restriction (IUGR) (RR 0.44, 95% CI 0.30–0.65) dropped if aspirin was included prior to 16 weeks’ gestation. The same researchers showed that aspirin introduced after 16 weeks of pregnancy does not reduce the incidence of PE (RR 0.81, 95% CI 0.63–1.03) or IUGR (RR 0.98, 95% CI 0.87–1.10) [46]. Similar conclusions regarding the timing of aspirin inclusion were reached by the authors of a meta-analysis involving 20,909 women, where ASA included before 16 weeks’ gestation was key to causing a decrease in the incidence of PE, while inclusion after 16 weeks’ gestation had no such effect. Additionally, they assessed that aspirin at doses of 100 and 150 mg was more effective than doses of <75 mg [17]. In most of the aforementioned meta-analyses, ASA was included in the treatment of women with risk factors for PE indicated by their medical history.

Undoubtedly, the ASPRE trial study was another breakthrough in PE prevention. It was a randomized analysis with a placebo group to evaluate aspirin use at a dose of 150 mg in women at high risk of PE. The novelty was that the risk group was identified in the first trimester based on a Fetal Medicine Foundation (FMF)-proposed algorithm including history, uterine artery pulsatility index (UtPI), mean arterial pressure (MAP), and PlGF [11,12,13]. In total, 1776 women with a risk of >1:100 were administered either aspirin or the placebo. The study proved a lower incidence of preterm PE (Odds ratio (OR) 0.38; 95% CI, 0.20–0.74) in the aspirin-taking group [19]. A secondary analysis of the ASPRE trial indicated an even greater decrease in PE incidence where ASA was taken regularly (>90% of the doses) in the aspirin-taking group (5/555) compared to the placebo (22/588) (OR 0.24, 95% CI 0.09–0.65). If hypertensive patients were excluded, this effect was actually spectacular in the aspirin-taking group (1/520) compared to the placebo (19/541) (OR 0.05, 95% CI 0.01–0.41) [47]. None of the aforementioned studies confirmed the benefits of aspirin use in the prevention of term preeclampsia [19,43,44,45,46,47]. Following the ASPRE trial, a number of papers have been published that evaluated aspirin use in the Asian and European populations based on the FMF screening model. Their authors showed that this screening, together with subsequent implementation of aspirin, was better than the screening methods proposed by, say, the National Institute for Health and Care Excellence (NICE) and others. For the FMF screening model, they showed a detection rate (DR) of 75–80% at a false positive rate (FPR) of 7–10%, while for maternal history alone the DR was approx. 35–40%. The researchers pointed out the need to choose the right cutoff point for the right population as, for example, in the Asian population, the screen-positive rate (SPR)was 23% at the recommended cutoff point of <1:100. An additional advantage of introducing such screening is that, as the authors emphasized, it is a method that identifies pregnancies that require appropriate monitoring due to higher risk of preterm labor, FGR, and the need for emergency termination of pregnancy [48,49,50].

Is aspirin effective in all cases? Unfortunately, as the authors show, some groups of women may not benefit as significantly. Situations worth mentioning are women with chronic hypertension and cases of multiple pregnancies. The authors of a meta-analysis involving 2150 women with chronic hypertension receiving LDA showed no significant statistical reduction in the incidence of preterm PE (OR 1.17, 95% CI 0.74–1.86) [51]. In addition, the aforementioned ASPRE secondary analysis showed no significantly reduced incidence of preterm PE among women with chronic hypertension in the aspirin-taking group (5/49) compared to the placebo (5/61) (OR 1.29, 95% CI 0.33–5.12) [47]. The authors speculate that the lack of a positive effect of LDA in this group is due to pre-pregnancy endothelial damage and inflammation, and PE develops even in less severe cases of abnormal placentation [47,51,52]. The second group that needs to be discussed are the aforementioned women with multiple pregnancies. The authors of a meta-analysis involving 898 multiple pregnancies receiving LDA observed a reduction in the risk of PE (RR 0.67, 95% CI 0.48–0.94) but not its severe forms (RR 1.02, 95% CI 0.61–1.72). In addition, this reduction did not differ when LDA was introduced before (RR 0.86, 95% CI 0.41–1.81) or after (RR 0.64; 95% CI 0.43–0.96, *p* = 0.50) 16 weeks of gestation [53]. Another meta-analysis involving 2273 multiple pregnancies showed a lower risk of PE among women receiving LDA (OR 0.64, 95% CI, 0.48–0.85). When they only evaluated 804 women receiving LDA at a dose of >100 mg/d, the risk was even lower (OR 0.45, 95% CI 0.23–0.86) [54]. The authors of these last two studies self-reported the low quality of the evidence and the need for further randomized trials showing the effectiveness of LDA in preventing PE among multiple pregnancies [53,54].

## 5. Is Aspirin Right for Every Woman?

ASA has been proven to be effective in preventing the development of PE. Hence the question of whether or not it would be simpler to include it in every pregnancy [55,56]. For years now, researchers have been attempting to prove that aspirin is safe to use. Clinicians treat it as safe to use in pregnancies. Nevertheless, according to the Food and Drug Administration (FDA), the use of aspirin at a dose of >81 mg in pregnancy continues to be an off-label indication [57]. Therefore, it should be explained in detail to every patient why higher doses must be used in her treatment whenever that is necessary. To date, no association of LDA with the development of birth defects, malformations, miscarriages, and premature closure of the ductus arteriosus has been detected [58,59,60,61]. One study evaluating the use of paracetamol, ibuprofen, and aspirin on 185,617 pregnant patients showed the possibility of increased risk of cerebral palsy in women exposed to aspirin during pregnancy. However, the paper did not report the exact dose, time of inclusion, or duration of aspirin treatment, as it only indicated whether ASA was used or not during pregnancy, and thus, this result should be approached with caution [62]. On the other hand, the authors of two studies on more than 10,000 cases showed no adverse effect of LDA on the neurodevelopment of children at 18 months compared to the group taking a placebo [63,64].

The use of LDA in pregnancy in relation to the incidence of bleeding, where the results are inconclusive, is a somewhat different story. On the one hand, one can find studies, such as the one on 26,952 patients, showing no association between aspirin administration and the incidence of postpartum hemorrhage (RR 1.03, 95% CI 0.94–1.23), placental abruption (RR 1.15, 95% CI 0.76–1.72), and neonatal intraventricular hemorrhage (RR 0.90, 95% CI 0.51–1.57) [65]. The ASPRE trial, too, did not show such a correlation, with the results in the placebo and aspirin groups being similar [19]. On the other hand, there are papers that point to the possible existence of such a relationship. Authors from Sweden explored the effects of LDA on the incidence of complications in 313,624 patients. They found no association with the incidence of midgestational bleeding. However, they made more diagnoses of intrapartum hemorrhage (adjusted odds ratio (aOR) 1.63, 95% CI 1.30–2.05), postpartum hemorrhage (aOR 1.23, 95% CI 1.08–1.39), and postpartum hematoma (aOR 2.21, 95% CI 1.13–4.34) and reported a higher incidence of intraventricular hemorrhage in newborns (aspirin: 0.07% vs. no aspirin: 0.01%; aOR, 9.66, 95% CI 1.88–49.48) born naturally. The authors admitted, though, that they did not know when their patients had stopped taking aspirin [66]. A meta-analysis by Cochrane, too, points to a low connection with the incidence of postpartum hemorrhage (*n* = 40,249, OR 1.06, 95% CI 1.00–1.12) [44]. A study on 21,403 patients claimed that the use of LDA was associated with increased risk of placental abruption (OR 1.35, 95% CI 1.05–1.73) [43]. Increased risk of hemorrhage may also be suggested by the fact that when a population of non-pregnant women with no elevated cardiovascular risk was studied, aspirin was shown to only increase the risk of external hemorrhages, gastrointestinal bleeding, and intracranial hemorrhage, without reducing the risk of, for example, myocardial infarction or ischemic stroke [67,68]. This is why, as the authors emphasized, LDA should be reserved for those at high risk of cardiovascular events in non-pregnant patients as well [67,68,69,70]. Therefore, the scientific societies’ recommendation that aspirin be discontinued in the 36th week of gestation seems reasonable as the time of delivery closes in, especially since no effect of LDA on the incidence of term PE has been demonstrated [71,72,73,74,75,76]. Additionally, the authors argue that such universal application might reduce aspirin compliance [77,78].

Is compliance really that important? There have been reports claiming that appropriate and regular use reduces the risk of various complications. High medication compliance has been linked, for example, to reduced mortality in depressed patients using antidepressants, reduced incidence of cardiovascular diseases in hypertensive patients using pharmacotherapy, and reduced risk of death in statin users with diabetes [79,80,81]. Is the level of adherence to a particular treatment common in the population? As the authors of a review paper covering 50 years of research report, approx. 25% of the population do not adhere to recommendations [82]. Pregnant patients demonstrate varying degrees of compliance as well. As regards recommendations for the use of vitamins or dietary suggestions, there are publications claiming that up to 70% of women do not follow one of the recommendations they receive [83]. The situation looked somewhat better among pregnant women with chronic diabetes mellitus, depression, or epilepsy, where adequate compliance was declared by 80–100% of them. However, the same authors indicated that compliance dropped significantly when it came to medications prescribed for a limited period, such as antibiotics, analgesics, steroids, or antihistamines (12–77%) [84]. Other authors studying aspirin compliance in pregnancy point to varied adherence as well. For women at high risk of PE, compliance ranged from approx. 50% to over 90% [85,86,87]. They also required to be reminded about their recommendations more often. In the case of the intermediate-risk patients, compliance declined, while physicians were rarely forced to remind their patients of their recommendations [85,86]. In the reported studies, the patients themselves emphasized that being reminded of the recommendations, as well as maintaining appropriate contact and cooperation with the medical staff, improved their aspirin compliance [88]. As we showed earlier, secondary analyses of the ASPRE trial found conclusively that patients taking more than 90% of their aspirin doses had reduced incidence of PE compared to those with lower compliance [47,87]. Other researchers studying high-risk pregnancies also show that compliance of <90% is associated with higher risk of early-onset preeclampsia (aOR 1.9, 95% CI 1.1–8.7) and higher risk of its late-onset form (>34 weeks’ gestation) (aOR 4.2, 95% CI 1.4–19.8). The authors did not study the compliance relationship for term PE [89].

Knowing how important compliance is and how many women fail to adhere to their prescribed pharmacological treatment, is it worth recommending extensive use of aspirin in pregnancy? The scientific societies, too, are in agreement as to this issue and discourage the universal use of aspirin across the population [3,71,72,73,74,75,76,90,91,92,93,94,95,96,97]. Women at increased risk of developing PE should be carefully identified within the general population on the basis of comprehensive first-trimester screening tests and/or their maternal and obstetric history [11,12,13,15].

## 6. What Are the Approaches to Screening for PE Worldwide?

For a long time, the only method to single out patients at high risk of developing PE was one based on the risk factors identified during early pregnancy. Numerous papers focusing on risk factors evaluating from 265,270 to as many as 25,356,688 pregnancies include history of preeclampsia (RR 8.4, 95% CI 7.1–9.9), chronic hypertension (RR 5.1, 95% CI 4.0–6.5), pre-gestational diabetes mellitus (RR 3.7, 95% CI 3.1–4.3), antiphospholipid syndrome (RR 2.8, 95% CI 1.8–4.3), systemic lupus erythematosus (RR 2.5, 95% CI 1.0–6.3), chronic kidney disease (OR 10.4, 95% CI 6.3–17.1), obesity (aOR 3.7, 95% CI 3.5–3.9), and family history of preeclampsia (RR 2.9, 95% CI 1.7–4.9). Others factors, which are equally significant, include multiple pregnancy (RR 2.9, 95% CI 2.6–3.1), primiparity (RR 2.1, 95% CI 1.9–2.4), use of assisted reproductive technology (ART) (RR 1.8, 95% CI 1.6–2.1), maternal age > 35 years (RR 1.2, 95% CI 1.1–1.3), black race (adjusted hazard ratio (aHR) 1.6, 95% CI 1.5–1.6), history of placental abruption (RR 2.0, 95% CI 1.4–2.7), and stillbirth (RR 2.4, 95% CI 1.7–3.4) [98,99,100,101]. We can even find a study evaluating socioeconomic status as a potential factor in the development of PE. The authors studied 3547 pregnant women, where after taking into account factors such as family history, material factors, psychosocial factors, substance use, working conditions and preexisting medical conditions, they showed a higher incidence of PE in the group with lower socioeconomic status (aOR 4.91, 95% CI 1.9–12.5) [102].

Apparently, the list of factors is long, and it has not been exhausted yet. Each of them has a different impact on the incidence of PE, so societies have divided them into high and moderate risk factors on which basis recommendations for ASA use are established. Table 1 summarizes the risk factors and differences in the statement of the various societies, which in the given recommendations are taken into account in identifying patients at high risk of PE. The societies divide them into those of high risk (red) and moderate risk (yellow), and some of them are not considered at all (gray) in risk estimation (see Table 1). Interestingly, several societies show a different approach to screening. Although they list risk factors in their recommendations (green color), they do not directly divide them into high or moderate risk factors and leave the decision to qualify them indirectly to the clinician depending on the screening method they choose (based only on risk factors or based on risk calculation according to FMF principles). Table 2 provides the actual indications for implementing LDA prophylaxis. The table shows that currently, according to the recommendations of societies such as the American College of Obstetricians and Gynecologists (ACOG), NICE, the American Heart Association (AHA), the European Society of Cardiology (ESC), the World Health Organization (WHO), and the Society of Obstetric Medicine Australia and New Zealand (SOMANZ), screening should be based solely on risk factors for PE when determining the indication for ASA [3,71,72,73,74,75,76,90,91,92,93,94,95,96,97]. This approach, however, may be fraught with poor detection of PE.

As the authors show, some studies regarding the use of ASA have shown using an algorithm according to NICE guidelines (based only on risk factors) that DR was only 40.8% for preeclampsia (PE) and 30.4% for all forms of PE with an FPR of 10.3% [103]. The DR for PE is completely different when using the algorithm proposed by FMF. The FMF assumed a different approach to the first-semester screening test for PE. Their researchers presented prospective studies evaluating maternal characteristics combined with a number of markers. Table 3 shows the most important studies for detecting PE at <32 weeks’ gestation, <34 weeks’ gestation, and before 37 weeks’ gestation [11,12,13,15].

Clearly, the last two of these algorithms has the highest DR, and the one without PAPP-A is the one that the FMF currently recommends as the algorithm of choice. If added to this algorithm, PAPP-A does not significantly increase the DR. In their evaluation of earlier-onset forms of preeclampsia (prior to 34 and prior to 32 weeks’ gestation), the same authors reported achieving an even higher DR of 89–100% while maintaining an FPR of 10% for the algorithm including maternal characteristics, MAP, UtPI, and PlGF [11,12,13,15]. They stressed that the DR might vary depending on the population studied; hence, it was extremely important that the appropriate cutoff point be chosen so that as many women as possible could benefit from ASA prophylaxis while maintaining a fairly low SPR. In their assessment, while using the above recommended algorithm for PE at < 37 weeks’ gestation, lower cutoff points could be more appropriate for black persons, as here the DR for a cutoff point <1:70 was approx. 88% at an SPR of approx. 25%, while for a cutoff point <1:100, it was 91% at an SPR of approx. 36%. As for Caucasians, it appears more reasonable to set the cutoff point at <1:100, where the DR is approx. 70% at an SPR of 7–11%, or <1:150, where the DR is 75–80% at an SPR of approx. 11–15% [11,12,13,15,104,105]. Due to financial and cultural constraints, it is not always possible to use complete screening for PE prior to 37 weeks’ gestation, which is why, for example, the International Federation of Gynaecology and Obstetrics (FIGO) and International Society of Ultrasound in Obstetrics and Gynecology (ISUOG) recommendations allow for using two-stage screening in which the first stage merely evaluates maternal characteristics and MAP, with UtPI and PlGF only assessed as an addition in cases of high risk [73,76].

The screening proposed by the FMF has been implemented successfully in a number of populations, and their results are promising, with their DRs reaching similar levels [106,107,108]. The ASPRE trial, too, evaluated the DR for PE, where—with account taken of aspirin’s effect—the DR was 77% for preterm PE and 43% for term PE at an FPR of 9.2% [109]. Several papers have been published that compare PE detection using the FMF algorithm with those based on the risk factors according to the NICE and the ACOG guidelines. A team of researchers from Asia have found that at an FPR of 20%, 75.8% of preterm PE cases were detected using the FMF algorithm and 54.6% using the ACOG one. At an FPR of 5%, the DR for preterm PE was 48.2% using those proposed by the FMF and 26.3% using the NICE guidelines [110]. Other researchers, too, have demonstrated the superiority of the FMF screening model over the ACOG and NICE ones in detecting preterm PE, where with the FMF algorithm the DR was 75% at an FPR of 10%, with the NICE algorithm it was 34% at an FPR of 10.2%, and with the ACOG model it was 5% at an FPR of 0.2%. When the ACOG recommendations for detecting high-risk patients were taken into account, the DR was indeed 90% for preterm PE, although this was at an FPR of 64.2% [111]. Here too, research has shown that in the case of the FMF screening test for PE prior to 37 weeks’ gestation, the DR was 74.8% at 10% SPR. In the same paper, when they used the ACOG recommendations, 89.2% of were detected, although at an SPR of 66.1%. As for the NICE algorithm, authors showed a DR of 42% at an SPR of 11.5% [13]. However, the last two of the aforementioned studies used ACOG’s former screening guidelines, while the current recommendations have been expanded slightly [65,97].

An up-to-date comparison between the NICE and FMF screening models has been depicted in a study in which the FPR was significantly lower with a significantly higher rate of aspirin prophylaxis in women who developed PE in the case of the FMF algorithm. In addition, the same authors showed what would happen after the introduction of full PE screening according to FMF with the subsequent use of aspirin, compared to the pre-intervention period where screening according to NICE was used. The number of observed cases of preterm PE could drop by as much as 80% over the next 21 months compared to standard screening according to the NICE model [48]. Summarizing these studies, it is manifest that depending on the screening algorithm used (the FMF vs the NICE vs the ACOG), a different proportion of women will be qualified for aspirin administration. This is inasmuch important as with the FMF and NICE algorithms this figure most often ranges between 10% and 20%, whereas with the ACOG recommendations up to 2/3 of the general population may be forced to take LDA. As we showed above, such a large number may have two extremely important implications:(1)It may reduce aspirin compliance;(2)It may cause more adverse reactions to such therapy in the population.

Taking the above into consideration, numerous scientific societies, such as ISSHP (Canada), Society of Obstetricians and Gynaecologists of Canada (SOGC), Brazilian Society of Hypertension (SBH), Brazilian Society of Nephrology (SBN), Polish Society of Hypertension (PSH), Polish Cardiac Society (PCS), Polish Society of Gynaecologists and Obstetricians (PTGiP), German Society of Gynaecology and Obstetrics (DGGG), Austrian Society of Gynecology and Obstetrics (OEGGG), Swiss Society of Gynecology and Obstetrics (SGGG), and Queensland Clinical Guidelines (Australia), recommend the FMF algorithm as the one of choice in first-semester screening for PE and only suggest resorting to risk factor-based evaluation if the former is unavailable (see Table 2) [3,71,72,75,91,96]. The FIGO and ISUOG, on the other hand, recommends that the FMF screening model be used as the algorithm of choice, and only if complete screening is not possible, then at least that an assessment of the risk factors, history, and MAP should be carried out [73,76]. The up-to-date recommendations as to the screening method, ASA dose, and its inclusion time are shown in Table 2. The scientific societies disagree as to the optimal screening method as well as aspirin dose and the time of its inclusion. However, taking a closer look at the societies’ current and previous recommendations collected in literature reviews [112,113], the following conclusions can be made:(1)In publications on aspirin dosing for the prevention of PE, the societies are inclined to recommend using increasingly higher doses (>75 mg) of ASA, with many encouraging doses from 100 mg up or a dose of 150 mg exclusively.(2)Most societies’ up-to-date recommendations indicate that the optimal time for implementing aspirin is prior to 16 weeks’ gestation, while some of the previous recommendations varied markedly in this regard.(3)Following the publication of the ASPRE trial results and other papers evaluating the FMF screening model, a large number of societies have changed their recommendations from those based on risk factors alone to the one based on the risk assessment proposed by the FMF for the first-semester screening test.

## 7. Conclusions

Globally, PE continues to be an extremely dangerous pregnancy complication, as an effective treatment has yet to be found. Nevertheless, the recent years have seen remarkable progress in the prediction and prevention of preterm forms of PE. It is evident that the previous approach to first-trimester risk factor-based screening for PE is being superseded by far more accurate methods that assess maternal factors and biophysical and biochemical measurements. This allows for detection of a high-risk pregnancy population in whom aspirin will be remarkably effective in preventing preterm PE. Such a strategy makes it possible to adequately monitor such pregnancies and reduce the overall risk related to the development of preeclampsia, thereby decreasing maternal and fetal morbidity and mortality worldwide.

## Figures and Tables

**Table 1 biomedicines-11-01495-t001:** Maternal risk factors for preeclampsia according to professional organizations.

Country/Spread of the Scientific Society	International	Canada	USA	USA	Brazil	Poland	UK	Europe	Switzerland	International	Australia, New Zealand	Australia	Germany, Austria, Switzerland
Society	ISSHP 2021	SOGC 2022	ACOG, SMFM, USPSTF 2021	AHA 2022	SBH, SBN 2020	PCS, PCH, PTGiP 2018	NICE 2019	ESC 2018	WHO 2011	FIGO 2019 ISUOG 2023	SOMANZ 2014	Queensland Clinical Guidelines 2021	DGGG, OEGGG, SGGG 2018
Previous pregnancy with preeclampsia													
Chronic hypertension													
Type 1 or type 2 diabetes mellitus													
Renal disease													
Multifetal gestation													
Antiphospholipid syndrome													
Systemic lupus erythematosus													
Hypertension in previous pregnancies													
Nulliparity													
Overweight (BMI > 25)													
Obesity (BMI > 30)													
BMI >35													
Family history of preeclampsia (mother or sister)													
Black race													
Lower income													
Age 35 years or older													
Age 40 years or older													
In vitro fertilization													
Low birth weight or small for gestational age													
Previous adverse pregnancy outcome													
>10-year pregnancy interval													
Previous FGR													
Previous placental abruption													
Previous stillbirth													
Systolic BP >130 mm Hg or diastolic BP >80 mm Hg before 20 wkGA													
Maternal congenital heart defects													
Maternal anxiety or depression													
Increased uterine artery resistance after 24 wkGA													

Legend: 
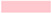
—high risk factor for PE; 
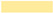
—moderate risk factor for PE; 
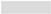
—not included in PE risk assess; 
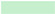
—included in PE risk assess; ISSHP: International Society for the Study of Hypertension in Pregnancy; SOGC: Society of Obstetricians and Gynecologists of Canada; ACOG: The American College of Obstetricians and Gynecologists; SMFM: Society for Maternal-Fetal Medicine; USPSTF: U.S. Preventive Services Task Force; AHA: American Heart Association; SBH: Brazilian Society of Hypertension; SBN: Brazilian Society of Nephrology; PCS: Polish Cardiac Society; PSH: Polish Society of Hypertension; PTGiP: Polish Society of Gynaecologists and Obstetricians; NICE: National Institute for Health and Care Excellence; ESC: European Society of Cardiology; WHO: World Health Organization; FIGO: The International Federation of Gynecology and Obstetrics; ISUOG: International Society of Ultrasound in Obstetrics and Gynecology; SOMANZ: Society of Obstetric Medicine of Australia and New Zealand; DGGG: German Society of Gynaecology and Obstetrics; OEGGG: Austrian Society of Gynecology and Obstetrics; SGGG: Swiss Society of Gynecology and Obstetrics; BMI: body mass index; BP: blood pressure; wkGA: weeks’ gestational age.

**Table 2 biomedicines-11-01495-t002:** Aspirin: preeclampsia screening test choice and risk-reducing recommendations by different societies.

Society	Method of Screening	Indication for Aspirin (ASA)		Dose of ASA	When ASA Should	
		First Choice	Second Choice		Start (Weeks)	End (Weeks)
ISSHP 2021	Preferred FMF screening Risk factors if FMF screening impossible	High risk from FMF screening	≥1 high risk factor or >1 moderate risk factor	150 when FMF used, 100–162 when from risk factors only	Before 16	36
SOGC 2022	Preferred FMF screening Risk factors if FMF screening impossible	High risk from FMF screening	≥1 high risk factor or >1 moderate risk factor	81–162	Before 16	36
ACOG, SMFM, USPSTF 2021	Risk factors only	≥1 high risk factor or >1 moderate risk factor	Not specified	81	12–28 (optimally before 16)	To delivery
AHA 2022	Risk factors only	≥1 high risk factor or >1 moderate risk factor	Not specified	Not specified (refers to ACOG)	12–28 (optimally before 16)	To delivery
SBH, SBN 2020	Preferred FMF screening Risk factors if FMF screening impossible	High risk from FMF screening	≥1 high risk factor or >1 moderate risk factor	75–150	Before 16	Not specified
PSH, PCS, PTGiP 2018	Preferred FMF screening Risk factors if FMF screening impossible	High risk from FMF screening (>1:150)	≥1 high risk factor or >1 moderate risk factor	100–150	Before 16	36
NICE 2019	Risk factors only	≥1 high risk factor or >1 moderate risk factor	Not specified	75–150	12	To delivery
ESC 2018	Risk factors only	≥1 high risk factor or >1 moderate risk factor	Not specified	100–150	12	36
WHO 2011	Risk factors only	≥1 high risk factor	Not specified	75	Before 20	Not specified
FIGO 2019	Preferred FMF screening; if full screening is impossible at least risk factors + MAP	High risk from full FMF screening (>1:100)	High risk from FMF screening (maternal characteristics + MAP)	150, when it is not possible 100 mg	11–14+6	36
SOMANZ 2014	Risk factors only	Moderate to high risk (Not differentiated between moderate and high risk factors)	Not specified	Low dose	Not specified	37
Queensland Clinical Guidelines 2021	FMF screening or risk factors	High risk from FMF screening	Moderate to high risk (Not differentiated between moderate and high risk factors)	100–150	Before 16	36
DGGG, OEGGG, SGGG 2018	Preferred FMF screening Risk factors if FMF screening impossible	High risk from FMF screening	Moderate to high risk (Not differentiated between moderate and high risk factors)	150	Before 16	34–36
ISUOG 2023	Preferred FMF screening, if full screening is impossible at least risk factors +MAP	High risk from full FMF screening (>1:100)	High risk from FMF screening (maternal characteristics +MAP)	150	11–15+6	36

Legend: ISSHP: International Society for the Study of Hypertension in Pregnancy; FMF: Fetal Medicine Foundation; MAP: Mean arterial pressure; SOGC: Society of Obstetricians and Gynecologists of Canada; ACOG: The American College of Obstetricians and Gynecologists; USPSTF: U.S. Preventive Services Task Force; SMFM: Society for Maternal-Fetal Medicine: AHA: American Heart Association; SBH: Brazilian Society of Hypertension; SBN: Brazilian Society of Nephrology; PSH: Polish Society of Hypertension; PCS: Polish Cardiac Society; PTGiP: Polish Society of Gynaecologists and Obstetricians; NICE: National Institute for Health and Care Excellence; ESC: European Society of Cardiology; WHO: World Health Organization; FIGO: The International Federation of Gynecology and Obstetrics; SOMANZ: Society of Obstetric Medicine of Australia and New Zealand; DGGG: German Society of Gynaecology and Obstetrics; OEGGG: Austrian Society of Gynecology and Obstetrics; SGGG: Swiss Society of Gynecology and Obstetrics; ISUOG: International Society of Ultrasound in Obstetrics and Gynecology; FMF: Fetal Medicine Foundation; MAP: Mean arterial pressure.

**Table 3 biomedicines-11-01495-t003:** Screening methods for preeclampsia according to research group.

Authors, Year	Akolekar et al., 2013 [15]	O’Gorman et al., 2016 [12]	O’Gorman et al., 2017 [11]	Tan et al., 2018 [13]
Study population	58,884	35,948	8,775	61,174
	DR for 10% FPR for PE < 34 weeks’ gestation	DR for 10% FPR for PE < 32 weeks’ gestation	DR for 10% FPR for PE < 32 weeks’ gestation	DR for 10% SPR for PE < 32 weeks’ gestation
Maternal characteristics plus:				
MAP	72.9	65	71	61.2
MAP + UtPI	89.7	80	94	82.8
MAP + UtPI + PAPP-A	92.5	83	94	82.8
MAP + UtPI + PAPP-A + PlGF	95.8	89	100	89.7
MAP + UtPI + PlGF	96.3	89	100	89.7
	DR for 10% FPR for PE < 37 weeks’ gestation	DR for 10% FPR for PE < 37 weeks’ gestation	DR for 10% FPR for PE < 37 weeks’ gestation	DR for 10% SPR for PE < 37 weeks’ gestation
Maternal characteristics plus:				
MAP	59.3	59	47	50.5
MAP + UtPI	71.5	70	71	68.4
MAP + UtPI + PAPP-A	74.6	70	69	68.2
MAP + UtPI + PAPP-A + PlGF	76.6	75	80	74.8
MAP + UtPI + PlGF	77.3	75	75	74.8

Legend: MAP: mean arterial pressure; UtPI: uterine artery pulsatility index; PAPP-A: pregnancy-associated plasma protein A; PLGF: placental growth factor; DR: detection rate; FPR: false positive rate; SPR: screen positive rate.

## Data Availability

Not applicable.

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
