# Peer review of "Low-Dose Aspirin after ASPRE—More Questions Than Answers? Current International Approach after PE Screening in the First Trimester"

_biomedicines, 2023, doi:10.3390/biomedicines11061495_

Round 1

Reviewer 1 Report

-----------------

General comments:

-----------------

[General] For the reviewer, if allowed, it would really help with inserted page and row numbers (Page number gets added some pages into the manuscript and hence gets faulty.)

[Display] If possible, to divide the text into more paragraphs. Section 3-5 have virtually not been divided in this sense at all. (To make the text less dense, etc.) 

[Literature] To find the relevant literature to base this review on, which literature search procedure was used here? (To add some further related notes in the beginning, i.e. before 'the actual review'?)

[Abbreviations] To explain (write out) all relevant and used abbreviations at first occurrence and then, as far as possible, use only the abbreviations after that point.

[Novelty] Perhaps to, at the end of Introduction, try to briefly summarize what has been written-and-summarized regarding this issue before and hence also give related notes on the aims-and-scope and aimed-for-novelty of this work, etc.?

[Tables] To give some more explanation in the main text of the included tables (what type of information they display) and also to note a bit more on their actual content (what information they actually give?

[Results] Perhaps to additionally aim at gathering all relevant numerical comparisons of methods (algorithms) at one place - e.g. in a table - where it might be easier to make a full comparison between methods (in the main text it gets a bit lengthy and potentially hard to follow)? 

------------------

Specific comments:

------------------

[X:Y; page X, row Y]

(Abstract)

1 As 'were compared'?

        Abbreviation 'ASA' not introduced.

        As 'FMF (Fetal...'? (add space)

(1. Introduction)

2:8 Abbreviation 'PE' introduced above.

2:18 As '30 years age.(10) Hence...'? (spacing, consistency) 

Also in 2:48, 2:51, 3:4, 3:23, 3:37, 4:7, 5:15, 5:44 and 7:6.

(2. An outline of...)

2:44 Sometimes space before reference(s), usually after full stop, sometimes not. Many occurrences throughout? Preferably to aim at consistency?

3:1-2 Abbreviation 'LDA' introduced above. Also in 3:6.

(3. What are the...)

3 Abbreviation to be introduced (written out) at first occurrence. Here for e.g. 'RR', 'CI', 'ASA'. Valid throughout.

3:21-23 I understand, but it doesn't read optimal. Some brief rephrasing?

3:22 End-parenthesis missing? (closing first results)

3:40 How many remaining women available for this analysis and, most importantly, how many observed events?

3:46-47 As indicated above, and valid throughout, explain introduced abbreviations at first occurrence.

(4. Is aspirin right...)

4:15 Other available trustworthy high-powered studies showing the opposite, i.e. no increased such risk? 

4:19-20 Here and onwards: Suggestively use result-display as used above, i.e. e.g. as '(OR 1.02, 95% CI 0.94-1.23)', also to avoid using double-parentheses. One may also consider to leave the '95% CI' label out for occurrences after the first case in any related result-presentation part, e.g. here 'OR 1.21, 0.96-2.07', etc.

Note: Isn't it strange that the CIs are so skew here? How were they calculated? In any non-standard way?

4:28 As 'newborns (aspirin: 0.07% vs. no aspirin: 0.01%; aOR 9.66, 1.88-49.48)...'?

4:31 As 'n=40,249'? (thousands-commas)

5:10-12 Was this tested formally? Formal significance? 

5:12-15 Why not give actual estimates and CIs also here?

5:14-15 Is the 'late-onset form' here equivalent to 'term PE' above? If so, does this not contradict the general findings as discussed above?

(5. What are the...)

5:24-34 References for all results? 

Brief info on corresponding studies (sample size, etc.)?

5:25 Missing closing-parenthesis. 

Note suggestion above on consistent layout for results-display.

5:26 As '2.8'? Also in 5:31 ('1.2'). (remove space)

5:32 No space after 'RR', i.e. as 'RR,'?

5:32-34 Underlying assumed factors?

5:48-52 Have all these (marker) abbreviations been introduced, and briefly explained, above? If not, it would make it more clear, I think.

Was the performance of these studies evaluated using independent test sets (or based on initial fitting [training] sets)?

5:53 It doesn't appear fully 'clear' that the last algorithm displays the best performance; the last two appear similar (given the presented information)?

6:1 Was this tested formally? 

6:1-4 Since both 32w and 34w are mentioned, perhaps to present results separately for these two distinct cases?

6:7 Abbreviations - as 'SPR' here - should be introduced at first occurrence. Does this refer to the 'screen-positive rate'? Or should it be 'FPR' here?  (Many such SPR-occurrences below among the FPR-occurrences.) 

6:25-27 Use consistent 'order' of results, e.g. first FMF, next others, throughout?

6:30-33 The ACOG comparison seems a bit off since their FPR turned out very low?

6:41-42 A bit unclear. 'Higher' compared to what?

6:42-44 The number of selected 'PE high-risk' cases or the actual number of observed cases? If the latter, more information on how this was found and if this was then 'caused' solely by aspirin use.

6:53-55 To avoid double-parentheses, perhaps as e.g. '(...; Poland: PSC, PCS, PTGiP; ...)'?

7:6 Introduce abbreviations - here 'MAP' - at first use/occasion.

(6. Conclusions)

10 No specific comments.

(References)

11-15 No specific comments.

(Tables)

7-8: Table 1: Figure title above table and with larger font size?

        What is the difference of the 'included' (green) parts compared to the 'high risk' (red) and 'moderate risk' (yellow) parts?

        All parts of the table (the right-most part) are not visible to me in the pdf made available to reviewers.

        All abbreviations among labels (left-most column) should preferably be explained in the legend below the table?

        'Country' is not a fully representative label (see e.g. 'Europe', 'International')?

        Keep order of organizations in legend (e.g. 'SMFM' before 'USPSTF')?

        As 'PCS' in header and as 'PSH' in legend?

        No 'ISUOG' in legend? Ok, added much later (see above).

        Columns 'DGGG'-'SGGG' not visible to me in the table (see above).

8-9: Table 2: Currently title on separate page than the actual table.

        Top: As 'Indication for aspirin (ASA)'? (space)  

        Vertical alignment of headers?

        See comments to Table 1 regarding e.g. order of organizations.

No specific comments.

Reviewer 2 Report

In my opinion this paper has a valuable content, discussing the still actual problem concerning LDA usage during pregnancy. Authors have shown different aspects of this problem, presenting actual knowledge of LDA action, and wide aspects of all controversies connected with Aspirin out- and during pregnancy. The problem of preeclampsia is also widely and properly described as well as the potential role of LDA in its prophylaxis.

Author Response

Thank you very much for your review. We hope this article bring significant improvement to our future scientific efforts.

Reviewer 3 Report

This article reviewed the methods of predicting preterm preeclampsia in first trimester and the methods of administering aspirin at various societies around the world, and discussed the effectiveness of aspirin in preventing preterm preeclampsia. This is potentially interesting paper. But this article should be changed in the following main points and minor points.

Main points

The preventive effect of aspirin on preterm PE in chronic hypertension and multiple pregnancy is still controversial and the authors should review and summarize it.

Minor points

Page 1, line 5 in Abstract: The word “acetylsalicylic acids” should be followed by the abbreviation “ASA” in parentheses.

Page 1, line 10 in Abstract: the Authors should initially use the word ”The Aspirin for Evidence-Based Preeclampsia Prevention” in full, followed by the abbreviation “ASPRE” in parentheses.

Page 1, line 10 in Abstract: the Authors should initially use the word ”Fetal Medicine Foundation” in full, followed by the abbreviation “FMF” in parentheses.

In main text: the Authors should initially use the words in full, followed by the abbreviation in parentheses for the following words (AKI, VEGF, PlGF, FGR, sFlt-1, RR, CI, ASA, ASPRE, FMF, OR, DR, FPR, SPR, FDA, HR, ACOG, AHA, ESC, WHO, SOMAZ, PAPP-A, 37Hbd, FIGO, ISUOG, SOGC, SBH, SBN, PSC, PCS, PTGiP, DGGG, OEGGG, and SGGG).

Table 1 and Table 2: The tables are cut off in the middle and I cannot see the whole tables. Please correct.

Page 9, in Table 2: Abbreviations of “FMF” and “MAP” should be specified in legend.

Round 2

Reviewer 1 Report

-------------------
Remaining comments:
------------------

[General] Many thanks to the authors for considering, and responding to, all my stated comments and questions.

[General] Check carefully also the new or edited parts with respect to e.g. consistent spacing (before start-parentheses, after end-parentheses, after references, etc.), usage of full stops (after citation marks ending sentences, etc.), having upper-case first letters in words only for true names or in the beginning of sentences, to introduce and write out all abbreviations at first occurrence (e.g. ASPRE), use of thousands commas (e.g. '2,273') throughout instead of forms as '10000' (without decimal) or '265.270' (using full stop), correct spelling (e.g. 'status' instead of 'stasus'), no double full stops (e.g. 'approx..'), that all sentences are efficiently, correctly and optimally worded and presented.  

Relatedly, see all previous comments on e.g. layout, notation and presentation (consistency, order), etc. 

No specific comments apart from the importance of also carefully checking the new or edited material with respect to this version (the updates of this draft). 

Author Response

Thank you very much for your review. We’ve corrected the text in accordance with your comments.

Reviewer 3 Report

This paper was precisely revised, it got better. But this article should be changed in the following minor points. If the minor points are changed, it would be worthy of accept.

Minor points

Page 2, line 79: The word “acetylsalicylic acids” should be changed by the abbreviation “ASA”.

Page 3, line 107: The word “fetal growth restriction” should be followed by the abbreviation “FGR” in parentheses.

Author Response

(The authors gave the same response as above.)
